# Differing terminology used to describe antimicrobial resistance can influence comprehension and subsequent behavioural intent

Kate Grailey [1] ✉, Ailidh Finlayson[2], Bobby Stuijfzand[2], Clare McCrudden[1], Adam Jones[2], Elena Meyer zu Brickwedde[2], Helen Brown[1], Sarah Huf[1], Hannah Behrendt[2] & Ara Darzi [1]

## Abstract

**Background** Despite global campaigns, the public's understanding of antimicrobial resistance (AMR) as a global emergency remains limited. Behaviour change is crucial in preserving antimicrobials but requires improved understanding of AMR at a population level. **Methods** Members of the public co-designed intervention arms, selecting three alternatives to AMR to be evaluated in a four-armed online randomised behavioural experiment. The primary outcome was attitudes towards AMR. Secondary and exploratory outcomes included comprehension, behavioural intent and recall. **Results** In April 2024, 4296 participants completed the online survey. *Antibiotic Resistance* is the most effective terminology for attitudes towards AMR ($p < 3.95E-06$), comprehension ($p = 0.013$) and recall ($p < 0.0003$). Both *Antibiotic Resistance* and *The Antibiotic Crisis* reduce behavioural intent to stop a course of antibiotics early. **Conclusions** Alternative terminology to describe AMR can impact attitudes, comprehension and behavioural intent towards antimicrobial use. Co-designing such terminology with the public can be an effective way utilising meaningful language in public health campaigns.

## Plain language summary

Antimicrobial resistance (AMR) occurs when microbes such as bacteria or viruses adapt and can no longer be treated with antimicrobials such as antibiotics. This threatens to become a global emergency in which simple infections cannot be treated unless we take widespread global action. Changing people's behaviour can help reduce the risk of AMR, if it enables appropriate prescribing of antibiotics or more people taking their antibiotics exactly as prescribed. The term AMR or antimicrobial resistance is not well understood by the public, so we worked with members of the public in London, U.K. to co-design alternative ways to provide more information about AMR, and tested the effect of these in an online survey. The terminology *Antibiotic Resistance* and *The Antibiotic Crisis* were found to be more effective in terms of understanding the importance of AMR, its consequences and inspiring people to use antibiotics more effectively. Our work suggest that using terms already familiar with the public such as 'antibiotics' could make public health campaigns on AMR more effective and memorable.

Antimicrobial resistance (AMR) is the ability of microorganisms, such as bacteria, viruses, fungi, and parasites, to resist the effects of antimicrobial agents, rendering these treatments ineffective. This phenomenon is a complex interplay of evolutionary pressure, genetic factors, and environmental conditions[1]. The overuse and misuse of antimicrobials in both human and animal healthcare has accelerated the emergence and dissemination of resistant strains[2]. Infections caused by resistant pathogens are often more difficult to treat, leading to increased morbidity, mortality, and healthcare costs[3]. It has been predicted that there will be up to 10 million deaths per year by 2050 if we do not mitigate this crisis[3]. AMR poses a

[1]The Fleming Initiative, Institute of Global Health Innovation, Imperial College London, London, UK. [2]The Behavioural Insights Team, London, UK. ✉e-mail: k.grailey18@imperial.ac.uk

significant global health threat, with implications for food security, economic development, and public health. To address this challenge, a multi-disciplinary approach is required, encompassing surveillance, infection prevention and control, antimicrobial stewardship, research and development of new antimicrobials, and public awareness[4–6].

Human behaviour is a significant contributor to the aetiology of the AMR crisis[7], including poor compliance with antimicrobial prescriptions, requesting/providing antimicrobials for illnesses where they are not indicated and the overuse of antimicrobials in food production. Significant and sustained changes in both individual and collective behaviour are needed to contribute to the mitigation of AMR. In accordance with the Theory of Reasoned Action and the Theory of Planned Behaviour, the likelihood of an individual engaging with health services in a way that promotes optimal and appropriate antimicrobial use is dependent upon their intentions, these being determined by attitudes towards the behaviour[8]. As such public awareness must extend beyond knowledge of AMR and be translated into definitive changes in behavioural intent and action. One mechanism of achieving this is through meaningful public engagement and involvement in research[9].

Global research has repeatedly demonstrated that current awareness and understanding of AMR in the general public is markedly lower than desired[10,11]. A survey in Nigeria demonstrated just 8.3% of respondents had good knowledge of AMR, and 76.6% felt 'powerless to stop it'[12]. There are also common misconceptions regarding the aetiology of AMR—that it results from changes in the human body, or the antimicrobials themselves[13,14].

The first step in creating attitudes towards antimicrobial use that reflect their need to be used cautiously and correctly (as a mechanism that subsequently initiates changes in behavioural intent and action) must be through improving awareness of this global health crisis and an improved understanding of its implications. It is well documented that the acronym 'AMR', and longer form 'AMR' are not reaching the public consciousness effectively. A 2015 World Health Organisation survey across 12 countries found only half had heard of AMR, and just 1/5 of AMR[15]. These findings were supported by a similar study in the United Kingdom[16].

Previous research has evaluated the term 'AMR'—finding that it doesn't possess three key criteria of a successful communication strategy—pronounceability, meaningfulness and specificity[17]. As such, it is perceived that 'AMR' is 'inconsistently used, difficult to pronounce and lacks meaning for lay audiences'. The Wellcome Trust undertook a large body of work in 2019, involving in-depth interviews with experts and message testing in seven countries worldwide. They developed five key principles for how to frame AMR in a way that is best received by the public[18]. This report also recommends drug-resistant infections as the term most easily understood by the public. However, in the 5 years since the publication of this report, drug-resistant infections has not become a commonly used term.

Further work has also postulated that the framing of AMR needs to be more inclusive of the wider determinants of AMR—including environmental and animal factors (such as the use of antibiotics in the food chain); and to develop AMR-related communications that are specific to local contexts[19].

A 2023 online survey investigated both memorability and risk association for the most frequent AMR-related health terms[20]. These were 'AMR', 'AMR', 'Drug-Resistant Infections', 'Antibiotic Resistance', 'Bacterial Resistance' and 'Superbugs'.

The study found that both 'AMR' and 'AMR' scored consistently low on these two parameters. 'Antibiotic resistance' performed best, with 'Drug-Resistant Infections' second. However, the authors conclude that none of the current options sufficiently motivate a change in antibiotic use. There have been repeated calls within existing academic literature calls for new names or ways to present AMR, with standardised terminology that is accessible to the public[17,20,21].

This study aims to build on the existing literature evaluating the effectiveness of different terminology used to describe AMR, by exploring its impact upon behavioural intent, attitudes towards and comprehension of AMR, as well as recall of the term itself. To our knowledge, this is the first evaluation of the impact of AMR-related terminology upon behavioural intent with regard to antimicrobial use, and the first that incorporates co-design methodology[22,23]. The study team worked with members of the public to co-design the intervention arms and broader study design. We hypothesised that co-designed terminology to represent AMR, created with the public, or existing AMR terminology that had marked public appeal, would resonate better, improving comprehension, recall, attitudes towards AMR and better influencing behavioural intent.

Our primary research objective was to identify which terminology used to represent AMR had the largest influence on attitudes towards AMR (whether participants acknowledged the severity and importance of AMR). The secondary objective was to identify which terminology used to represent AMR is most likely to improve comprehension of the concept. There were three exploratory objectives, to investigate which terminology used to represent AMR has the best recall from participants; which term most influenced behavioural intent—particularly with respect to future antibiotic use; and to explore the differing impact of different terminologies used to represent AMR on population sub-characteristics, including age, ethnicity, and previous familiarity with the term AMR.

Our findings demonstrate that alternative terminology for AMR, such as 'The Antibiotic Crisis' or 'Antibiotic Resistance' can be more effective at conveying at conveying the importance of AMR as a global issue, understanding of the topic, and changing reported behaviours such as finishing a course of antibiotics. Notably, 'Superbugs', the third terminology tested demonstrates poorer recall than AMR or the other alternative terms.

## Methods
### Study design
This four-arm, parallel, online randomised behavioural experiment evaluated the impact of different terminology used to represent AMR on attitudes towards AMR, comprehension, behavioural intent and recall. The study was delivered online in the UK using *Predictiv*, a platform for running behavioural experiments built by the Behavioural Insights Team[24]. Ethical approval for the study was obtained from Imperial College London Research Governance and Integrity Team on the 21st March 2024 (6995049). The study was registered on clinicaltrials.gov (NCT06356285) on the 4th April 2024.

### Interventions
The control arm for the study was 'Antimicrobial Resistance - AMR'. The three intervention arms were determined through a process of public engagement and co-design work. The first stage of this was public engagement, in which workshops and stalls at community events were held, asking members of the public what AMR means to them, and challenging them to come up with suggestions for how to best present AMR to the public. Three focus groups were also held by the study team, each 60 min long and with a diverse range of public participants in terms of age, ethnicity and professional background. Insights from members of the public on AMR were documented, collated and reviewed by the study team to identify any key themes that could be taken forwards to the co-design stage. All suggested terminology for AMR were complied, duplicates removed and taken forward for review at the co-design workshops.

A diverse group of members of the public representing a range of ages, ethnicities and professional backgrounds were recruited from North West London to participate in the co-design process. Demographic data was not formally collected from participants. Three co-design workshops were held in local community venues, each lasting 60 min, and each attended by six members of the public. Three of these participants were consistent throughout all three workshops. The first session involved playback of the insights gathered from the prior public engagement work, an overview of what AMR is, and an open forum to suggest alternative terminology to describe AMR, supplementing the list generated in the public engagement work. The second co-design session involved the shortlisting of ideas down to a list of ten, considering which terms were felt to have a broad appeal and

convey understanding to a general audience. This shortlist was shared with all members of the public who had shown interest in the project asking them to vote for their top choices, and they were invited to share it with family and friends. The final co-design session involved selection of the final three interventions, and co-design of the visual image used to present the AMR-related terminology to study participants, in terms of poster design and associated text.

Of the final three intervention arms, two were known alternative terms for AMR that have performed adequately in previous work—'Superbugs' and 'Antibiotic Resistance', and a third 'The Antibiotic Crisis' was a more novel terminology for AMR selected through the co-design process. Inclusion of 'Superbugs' and 'Antibiotic Resistance' allowed our more novel arm to be trialled against the current 'next best' terminology, as identified in previous work[20].

### Participants

To be eligible for inclusion in this study, participants had to be resident in the United Kingdom, be aged over 18, have previously signed up to a market research panel and passed an attention check provided at the start of the survey. Participants were recruited using demographic quotas to ensure a representative sample of the UK population according to age, gender and ethnicity. The online Predictiv survey was shared with Cint[25], a panel aggregator, who provided up to 40 market research panel companies with the online link. Cint provides access to over 500,000 participants who have completed an online survey in the preceding 12 months, roughly representative of the UK population.

Participants were incentivised to participate in the study with a small financial incentive, in the region of £0.70 per participant (with some variation within a small margin depending upon recruitment quotas).

### Study delivery

An online survey was developed by the study team, initially providing participants with information about the study, the option to participate, a brief attention check (a simple question to ensure they have read and understood the question. If this was not answered correctly, a second attention check was performed). Participants were then randomly allocated into one of four arms in a 1:1:1:1 ratio. This randomisation process was computer generated and conducted within the Predictiv platform at the individual level, and was not weighted according to demographics or prior knowledge of AMR. Participants were blinded to the study arm they were assigned to. Continuing with the survey after reading the participant information sheet was taken as implied consent.

Following randomisation, participants were presented with a visual image, containing some simple information explaining AMR, a statement reflecting its urgency and a call to action to use antibiotics only when needed. Only the terminology used for the title of the poster changed in each arm ('Antimicrobial Resistance-AMR'/'Superbugs'/'Antibiotic Resistance'/'The Antibiotic Crisis'), ensuring that the impact of the terminology used to name AMR was the focus of the investigation. Participants were free to look at the image for as long as required. The four arms can be viewed in Fig. 1.

Participants were subsequently presented with a series of questions developed by the study team, designed to answer the research objectives: evaluation of attitudes towards AMR, comprehension of the subject, behavioural intent with respect to antibiotics (focusing upon likelihood to request antibiotics for an upper respiratory tract infection, preferred healthcare services utilised and future intent to finish a course of antibiotics), along with recall of the terminology. The full survey with detail regarding each survey item, can be viewed in Supplementary Methods.

Recall was measured in three ways. Firstly, by asking participants to select which (from a series of potential options) terminology they had seen at the top of the poster; secondly, to select from four potential options what the content of the poster was, and finally a free text box asking participants to describe AMR. This question was shown immediately after the poster, to ensure the answer was not affected by information presented in subsequent survey questions. Responses to the first two questions were binary, depending upon whether the participant's answer was correct or not, and summated across all participants to indicate the percentage of participants who correctly remembered both pieces of information.

To evaluate attitudes towards AMR, five statements relating to the severity and importance of AMR were presented to the participants, asking the extent to which they agreed with each one. Overall attitudes towards AMR were calculated as an overall sum of how many statements the participants moderately or strongly agreed with, with higher scores indicating more perceived severity of AMR.

Comprehension of AMR was explored by asking participants three multiple choice questions about the definition, causes, and solutions of AMR. The comprehension score was a percentage score of how many correct responses were given. These answers were offered in multiple choice format with incorrect responses as the other options. Participants were also asked to rate their confidence in the answers provided to the comprehension questions, rating themselves from 'extremely confident' down to 'not confident' on a four-part Likert scale.

Two measures of behavioural intent relating to antimicrobial stewardship were evaluated over two survey items. The first presented the participant with a scenario of a simple viral illness (cough, coryza and sore throat), and asked participants to rate the likelihood of seeking help from four different healthcare scenarios offered, including visiting a GP, visiting a pharmacist, requesting antibiotics or taking other medications. This was

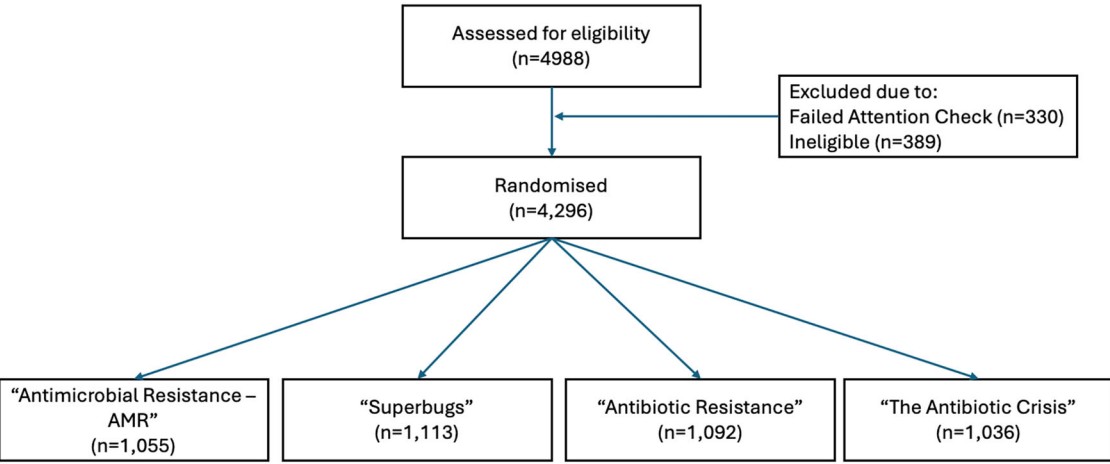

**Fig. 1 | Visual images as presented in each arm of the online randomised behavioural experiment.** The four images presented to participants in each study arm. Each poster has the same content, with only the terminology used to describe antimicrobial resistance differing.

followed by a second question posed to participants who stated they would request antibiotics in response to the first scenario, to investigate the likelihood of stopping a course of antibiotics early. Participants were asked to rate this likelihood on a 4-part Likert scale.

The survey also included some free text responses to further evaluate participants' understanding of AMR.

## Main measures

The primary outcome measure was the overall sum of binary ratings of five questions designed to evaluate attitudes towards AMR. The secondary outcome measure was the number of correct answers to three questions designed to investigate comprehension of AMR. Exploratory outcomes were measured as a binary Yes/No to the statements presented.

## Sample size

Accepting a two-sided $p$ value of 0.017, which is a $p$ value of 0.05 adjusted for 3 comparisons, as statistically significant, the study was designed with 80% power to detect a minimal detectable effect size of a 6% change between the study arms and control for the primary outcome. This was determined to require 1000 participants per arm. This is based upon a likely outcome baseline of 60% for the primary outcome, attitudes towards AMR, as referenced in previous similar studies by Wellcome[18], and a likely difference between trial arms as seen in other online studies[20].

Power calculations were conducted on a range of assumptions about the baseline of the primary outcome, and power ranged from a minimal detectable effect size between 3% and 7% as the outcome baseline changed from 80% to 50%.

## Statistics and reproducibility

For all outcomes, regression models were used to test each treatment arm against the control. All models were adjusted for the following covariates: age, gender, income, region, ethnicity, education, employment status, whether the participant prescribes antibiotics in their job, first language, and whether the participant has taken antibiotics in the last year. For the analysis of the primary and secondary measures, quasibinomial models were used as the outcomes are bounded count variables. For the binary exploratory outcomes, logistic regression models were used.

Exploratory subgroup analysis was conducted on the overall 'attitudes towards AMR' score for the following subgroups: existing familiarity with the concept of AMR; whether they had taken antibiotics in the past year; age; and ethnicity. Subgroup analysis was conducted by subsetting the data and then within each subgroup analysing the effect of treatment compared to control on the primary outcome using a quasibinomial regression.

For the primary and secondary outcomes separately, the Benjamini–Hochberg procedure was used to adjust for 3 comparisons (all treatment arms to control). There was no multiple comparison correction for the exploratory outcomes.

For all results, the raw control group mean is presented alongside adjusted means for each treatment group, which represent the control group mean adjusted for the treatment effect from the regression model. $P$ values are adjusted for primary and secondary outcomes, but unadjusted for exploratory outcomes. All confidence intervals reported are unadjusted.

## Free text analysis

Free text responses were analysed using a content analysis approach within Microsoft Excel, identifying common themes within the qualitative data.

## Reporting summary

Further information on research design is available in the Nature Portfolio Reporting Summary linked to this article.

# Results

## Participant characteristics

Recruitment to the online experiment opened on the 10th April 2024 and closed on the 18th April, by which point 4296 participants that represented the general UK population had completed the survey. Participant demographics can be viewed in Table 1. The median time spent completing the survey was 5 min 21 s. The flow of participants through the study is presented in Fig. 2. No harms were observed. De-identified data sets can be found at the project's Open Science Framework page[26].

## Primary outcome—attitudes towards AMR

Participants were asked to state their agreement with five statements relating to AMR. Participants in all four arms scored highly (all over 80%), indicating a good baseline presence of knowledge of the severity and importance of AMR. All three intervention arms used terminology to name AMR that outperformed 'Antimicrobial Resistance - AMR'. The highest performer was 'Antibiotic Resistance' (85%, $p = 3.95\text{E-06}$). When analysed according to individual statements, only 'Antibiotic Resistance' outperformed 'Antimicrobial Resistance—AMR' on all five. A subgroup analysis evaluated any differences in attitudes towards AMR according to age, ethnicity and previous stated familiarity with the concept of AMR. Of note, much higher scores were seen in older age groups (40–54 and 55+), and in both these age groups 'Antibiotic Resistance' and 'The Antibiotic Crisis' significantly outperformed 'Antimicrobial Resistance - AMR' (40–54 'Antibiotic Resistance': $p = 0.043$; 40–54 'The Antibiotic Crisis': $p = 0.030$; 55+ 'Antibiotic Resistance': $p = 0.0003$; 55+ 'The Antibiotic Crisis': $p = 0.026$). Most participants were familiar with the concept of AMR prior to the survey—which demonstrated a positive correlation between attitudes towards the severity and importance of AMR. Results can be viewed in Tables 2 and 3. Full logistic regressions for all models can be viewed in the Supplementary Data.

## Secondary outcome—comprehension of AMR

Participants were asked three multiple choice questions about the definition, causes and solutions of AMR. The effect size between the four arms was small, with a 3% difference between the highest performing arm—'Antibiotic Resistance' (on average 66.0% of answers correct, $p = 0.013$) and the lowest—'Antimicrobial Resistance—AMR' (63.3%, $p = 0.019$). Participants were also asked to rate their confidence in their answers to these comprehension questions, with confidence being over 95% for all four arms. Participants in the 'Antibiotic Resistance' arm were significantly more likely to be confident in their answers ($p = 0.018$), although the effect size was extremely small (1.2 percentage points) (Table 4).

## Exploratory outcomes—behavioural intent and recall

None of the intervention terminologies for AMR significantly reduced intent to request antibiotics when presented with symptoms of an upper respiratory tract infection. Both 'Superbugs' and 'Antibiotic Crisis' reduced intent to visit a general practitioner by three percentage points, and for the former this effect was significant ($p = 0.028$). There was no significant difference in intention to visit a pharmacist or take other medications between trial arms. Both the 'Antibiotic Resistance' and 'The Antibiotic Crisis' arms reduced intention to stop a course of antibiotics early by 13.4 and 15.4 percentage points respectively, the latter reaching statistical significance ($p = 0.050$). 'Antibiotic Resistance' showed significantly higher recall than AMR, with 7 percentage points more participants remembering the term ($p = 0.0003$). 'Superbugs' demonstrated worse recall, down 6.5 percentage points to 65.4% ($p = 0.001$) (Table 3).

## Content analysis

Over the two free-text questions, 8592 responses were recorded across all participants. A total of 565 responses contained keywords or phrases that indicated a correct understanding of AMR, such as 'bacteria are becoming more resistant to antibiotics'. Nine hundred and twenty-four (10.75%) participants explicitly stated that they did not know what AMR was including 'no idea' and 'nothing, I wasn't aware this is what this is'. Several misconceptions involving the aetiology of AMR were present, including that it was the human body or immune system which had changed so that antibiotics were no longer effective 'the human body becomes resistant to certain treatments/medications due to overuse or natural build up of

**Table 1 | Demographic data of participants recruited into the study**

| Demographic | | Percentage within 'Antimicrobial Resistance -AMR' group (*n* = 1055) | Percentage within 'Superbugs' group (*n* = 1113) | Percentage within 'Antibiotic Resistance' group (*n* = 1092) | Percentage within 'The Antibiotic Crisis' group (*n* = 1036) |
|---|---|---|---|---|---|
| Gender | Male | 54% | 52% | 50% | 52% |
| | Female | 46% | 47% | 50% | 48% |
| | Other | 0% | 1% | 1% | 0% |
| Age | 18–24 | 11% | 12% | 10% | 12% |
| | 25–39 | 26% | 25% | 25% | 25% |
| | 40–54 | 27% | 27% | 27% | 29% |
| | 55+ | 35% | 36% | 38% | 35% |
| U.K Region | South & East | 36% | 34% | 32% | 34% |
| | North | 20% | 21% | 23% | 24% |
| | Midlands | 16% | 16% | 17% | 15% |
| | Scotland | 7% | 8% | 8% | 7% |
| | Northern Ireland | 2% | 2% | 3% | 2% |
| | Wales | 5% | 4% | 6% | 5% |
| | London | 14% | 14% | 12% | 13% |
| Ethnicity | White | 85% | 83% | 86% | 85% |
| | Asian (Asian British) | 9% | 9% | 7% | 8% |
| | Black (or Black British) | 3% | 3% | 4% | 2% |
| | Mixed/Other | 4% | 5% | 3% | 4% |
| Education | Bachelors Degree | 33% | 33% | 30% | 29% |
| | No degree | 65% | 65% | 68% | 68% |
| | None of the above/ Prefer not to answer | 2% | 2% | 2% | 2% |
| Employment | Employed | 66% | 68% | 64% | 70% |
| | Inactive | 31% | 29% | 32% | 27% |
| | Unemployed | 3% | 3% | 3% | 3% |
| First language | English | 95% | 94% | 95% | 94% |
| | Other | 5% | 6% | 5% | 6% |
| Taken antibiotics in the past year | Yes | 34% | 37% | 34% | 35% |
| | No | 66% | 63% | 66% | 65% |
| Works in healthcare | Yes | 9% | 9% | 10% | 11% |
| | No | 91% | 91% | 90% | 89% |
| Prescribes antibiotics in job | Yes | 2% | 2% | 2% | 3% |
| | No | 98% | 98% | 98% | 97% |
| Pre-tax household income | Less than £40,000 | 46% | 45% | 48% | 44% |
| | £40,000 and over | 54% | 55% | 52% | 56% |

tolerance'. The top three categories of causes of AMR listed were; (1) Overuse/overprescription 'overuse of antibiotics has meant bacteria has changed to resist antibiotics. Some bacteria are now highly resistant'; (2) the natural evolution of microbes 'bugs that can adapt to changes' and (3) misuse/inappropriate use 'overuse, unnecessary consumption'. A small minority (0.2%) linked viral resistance to antibiotics 'the virus becomes immune to antibiotics'.

## Discussion

This online randomised behavioural experiment demonstrates the importance of the terminology used to describe AMR, with a clear impact upon behavioural intent for future antimicrobial use. Our study highlights that using the terms 'Antibiotic Resistance' and 'The Antibiotic Crisis' led to more participants being likely to report that they would finish a course of antibiotics—a key behaviour in mitigating the development of AMR. We also demonstrate that 'Antimicrobial Resistance - AMR' was outperformed

by 'Antibiotic Resistance' on attitudes towards AMR, comprehension of the topic and recall.

Our participant population demonstrated a high level of appreciation for the importance and severity of AMR. The terminology used to describe AMR did have an influence on the extent of this, with both 'Antibiotic Resistance' and 'Antibiotic Crisis' significantly increasing participants' perception of AMR as a serious global health problem. Of note, some population subgroups appeared to have a lower perception of AMR as a global health issue—those who were in the younger age groups, and those who had not previously been aware of AMR. This suggests that increasing exposure to antibiotics and healthcare services across a lifetime may have an influence on how important AMR appears to individuals. It may also be that a younger audience has a different perception of health threats in general, as they are typically less often affected by them.

Within our study, participants' comprehension of AMR (its definition, causes and potential solutions) was slightly influenced by the terminology

**Fig. 2 | CONSORT diagram.** CONSORT diagram illustrating flow through the online randomised behavioural experiment.

used to describe AMR, with participants in the 'Antibiotic Resistance' arm scoring significantly higher, however the effect size was small (2.7%). Between 60–65% of questions on AMR were answered correctly, however when asked to rate their confidence in their answers, participants rated this extremely highly (98–99%). This suggests that individuals may have strongly held false/misinformed beliefs about AMR, highlighting that policy makers and those designing public health communications should work to improve public understanding, make messaging clear and address any misinformation.

Interestingly, despite widespread media use in the United Kingdom, and scoring highly in our public engagement and co-design work, 'Superbugs' was the worst performer in terms of recall of the terminology, being significantly lower than 'Antimicrobial Resistance - AMR'. Again, the two terms with 'Antibiotic' incorporated performed better.

Semantic similarity has been shown to influence recall[27]—it may be that using the term antibiotics more explicitly within the terminology used to describe AMR may lead to stronger connections being built between its meaning, understanding and behavioural intent. Meaningfulness, familiarity and word length are also important[27,28], which may also contribute to the increased recall seen with these terms. Performance on recall also matched that on attitudes to AMR domains—with 'Antibiotic Resistance' demonstrating the highest scores for both. It is possible that this reflects the relationship between perceived importance of an issue and its subsequent memorability.

Given antibiotic use is the major driver of AMR[29], it follows that antibiotic stewardship must be a priority focus when striving to create meaningful public understanding and behaviour change. As such, our study focused upon exploring behavioural intent to use antibiotics, supported by our public engagement work and co-design process. Four measures of behavioural intent were explored, with mixed results. Using the terminology of 'Antibiotic Resistance' or 'The Antibiotic Crisis' markedly reduced the chance of a participant reporting that they would stop their course of antibiotics early—suggesting that by explicitly naming antibiotics as affected by AMR has an impact on their appropriate use, and may improve antibiotic stewardship. We did not show any impact on intent to request antibiotics with any of the terminologies presented, which may have been influenced by the call to action seen on the visual image itself, asking participants not to request antibiotics. Whilst the images were not designed to be educational, they included basic information about AMR which could have influenced a participant's intent. We did note that there was a significant reduction in the intention to see a Doctor (General practitioner) by approximately three percentage points for 'Superbugs' and 'The Antibiotic Crisis'. The reasons for this require further qualitative evaluation to understand fully, but may reflect the understanding that these simple respiratory illnesses seldom require antibiotics, and therefore can be manged without seeing a doctor. It

is also interesting to note that the terminology used to describe AMR does seem to influence the type of help sought, which needs to be interpreted with caution and explored further—as there is also a risk that fear of AMR may lead to less visits to healthcare professionals when needed.

This work supports existing literature describing Antimicrobial Resistance/AMR as poorly resonant with the general population[17,20,21], and highlights that there are terms that perform better, particularly with respect to intention to use antibiotics correctly, comprehension, attitudes towards AMR and recall. This study also supports the assertion in Wellcome's Reframing resistance that 'Superbugs' should be used sparingly, despite its widespread media popularity[19], given it performed worse than 'Antimicrobial Resistance - AMR' in terms of recall. Our study corroborates the findings in Krokow et al.'s 2023 study[20], where 'Antibiotic Resistance' also outperformed 'Antimicrobial Resistance' or 'AMR' in terms of evoking risk perception and memorability. Interestingly, 'Superbugs' demonstrated better memorability than AMR in that study, a finding which was not replicated in our work. This may be a reflection of how the terminology is presented, and provides an opportunity for future work in terms of understanding the effect of how AMR related terminology is presented, and not just the terminology itself.

To communicate effectively for action, we must ensure that information presented to the public is memorable and actionable. Both 'Antibiotic resistance' and 'The Antibiotic Crisis' are more closely linked to the most frequently desired call to action—to promote antibiotic stewardship and hence mitigate resistance. If we can turn the tide on antibiotic use, this may be the step that paves the way for a broader understanding of AMR by the public in time. Simplicity is acknowledged as important in conveying healthcare messages—something which is currently lacking in the term 'AMR'[30]. 'Antibiotic Resistance' and 'The Antibiotic Crisis' also house embedded triggers—by creating an explicit link to antibiotic use, the next time a patient requires antibiotics, they may be more likely to remember the concept of AMR and therefore use them appropriately[31,32]. Responses to the free-text questions, whilst demonstrating an overall good understanding of AMR, demonstrated the presence of some misconceptions, and a small majority having no knowledge of the subject, despite being presented with information during the study. This further emphasises the need for ongoing public engagement and awareness building, with focused clear information.

This study also showcases the benefits of extensive public involvement in co-creating the intervention arms and study design with the research team. We believe this adds to the robustness of our findings, when considering them as evidence that would support a widespread change in the focus of public health messaging on AMR. Adopting this process of co-design and user centred design throughout our study (including protocol development, engagement workshops, iterating and refining trial arms and design of the participant pathway), helped ensure the study evaluated

**Table 2 | Results of primary outcome—the impact of different terminology used to name antimicrobial resistance upon participants attitudes towards it as a global crisis**

| Outcome | | Terminology used to name Antimicrobial Resistance (AMR) | | | | |
| --- | --- | --- | --- | --- | --- | --- |
| | | Antimicrobial Resistance—AMR (Control) *Participant score* (%), *p/CI* | Superbugs *Participant score* (%), *p/CI* | Antibiotic Resistance *Participant score* (%), *p/CI* | The Antibiotic Crisis *Participant score* (%), *p/CI* | |
| **Primary Outcome** | | | | | | |
| Attitudes towards AMR (Overall)[a] | | 81.0% | 83.2% p = 0.066 CI = [80.8%, 85.3%] | 86.3% p = 0.00001 CI = [84.2%, 88.2%] | 85.2% p = 0.0006 CI = [82.9%, 87.1%] | |
| Attitudes towards AMR (individual statements) *AMR* replaced by terminology being tested in each arm. | *AMR** poses a risk to global health | 79.0% | 84.2% p = 0.002 CI = [81.0%, 86.9%] | 84.9% p = 0.0004, CI = [81.8%, 87.6%] | 83.7% p = 0.007, CI = [80.4%, 86.5%] | |
| | It is important to address *AMR* | 88.5% | 88.6% p = 0.986, CI = [85.5%, 91.0%] | 91.2% p = 0.043, CI = [88.6%, 93.3%] | 91.0%, p = 0.072, CI = [88.3%, 93.1%] | |
| | *AMR* is an urgent issue | 80.3% | 80.2% p = 0.958 CI = [76.5%, 83.4%] | 85.0% p = 0.005, CI = [81.8%, 87.7%] | 86.6% p = 0.0002, CI = [83.5%, 89.1%] | |
| | I understand what *AMR* means | 84.9% | 82.8% p = 0.183 CI = [79.2%, 85.9%] | 90.5% p = 0.0001, CI = [88.0%, 92.6%] | 89.9% p = 0.001, CI = [87.2%, 92.1%] | |
| | *AMR* is an issue that could impact my own health | 72.2% | 80.0% p = 0.00003, CI = [76.6%, 83.1%] | 79.9% p = 0.00004, CI = [76.5%, 83.0%] | 74.8% p = 0.197, CI = [70.9%, 78.3%] | |

[a]*p* values in this row are adjusted for multiple comparisons using the Benjamini–Hochberg procedure. CIs are unadjusted.

**Table 3 | Results of primary outcome—the impact of different terminology used to name antimicrobial resistance upon participants attitudes towards it as a global crisis according population subgroups**

| Outcome | | | Terminology used to name Antimicrobial Resistance (AMR) | | | |
|---|---|---|---|---|---|---|
| | | | Antimicrobial Resistance—AMR (Control) *Participant score (%), p/CI* | Superbugs *Participant score (%), p/CI* | Antibiotic Resistance *Participant score (%), p/CI* | The Antibiotic Crisis *Participant score (%), p/CI* |
| **Primary Outcome** | | | | | | |
| Variations in attitudes towards AMR across subgroups | Age | 18–24 (n = 487) | 73.9% | 73.3% $p = 0.874$, CI = [65.1%, 80.1%] | 79.3% $p = 0.165$, CI = [71.4%, 85.5%] | 77.4% $p = 0.354$, CI = [69.6%, 83.6%] |
| | | 25–39 (n = 1090) | 75.3% | 77.2% $p = 0.483$, CI = [71.8%, 81.8%] | 79.3% $p = 0.129$, CI = [74.1%, 83.7%] | 79.5% $p = 0.112$, CI = [74.3%, 83.9%] |
| | | 40–54 (n = 1178) | 83.2% | 84.9% $p = 0.422$, CI = [80.4%, 88.6%] | 87.5% $p = 0.043$, CI = [83.3%, 90.8%] | 87.8% $p = 0.030$, CI = [83.7%, 91.0%] |
| | | 55+ (n = 1541) | 85.9% | 89.1% $p = 0.057$, CI = [85.8%, 91.8%] | 91.8% $p = 0.0003$, CI = [88.9%, 93.9%] | 89.7% $p = 0.026$, CI = [86.4%, 92.3%] |
| | Ethnicity | White (n = 3664) | 81.5% | 83.9% $p = 0.071$, CI = [81.3%, 86.1%] | 87.0% $p = 0.00001$, CI = [84.8%, 88.9%] | 85.3% $p = 0.003$, CI = [82.9%, 87.5%] |
| | | Non-white (n = 652) | 77.9% | 78.8% $p = 0.771$, CI = [72.2%, 84.2%] | 81.3% $p = 0.287$, CI = [74.7%, 86.6%] | 84.9% $p = 0.024$, CI = [78/9%, 89.4%] |
| | Familiarity with concept | Not familiar (n = 594) | 59.2% | 68.1% $p = 0.062$, CI = [58.8%, 76.1%] | 76.6% $p = 0.000003$, CI = [68.2%, 83.3%] | 72% $p = 0.016$, CI = [61.9%, 80.4] |
| | | Familiar (n = 3702) | 82.7% | 86.5% $p = 0.002$, CI = [84.2%, 88.4%] | 88.2% $p = 0.0002$, CI = [86.1%, 90%] | 86.6% $p = 0.001$, CI = [84.4%, 88.5%] |

terminology already deemed to be the most meaningful to members of the public, in a way that resonates with their existing beliefs.

There are several limitations within this work. Due to the nature of recruitment, our participant sample does not capture those who are digitally excluded, or those not inclined to complete online surveys. Whilst some significant shifts in behavioural intent were demonstrated when using 'The Antibiotic Crisis' or 'Antibiotic Resistance' as the favoured terminology, particularly with respect to completing a course of antibiotics, this may not reflect real world action. This provides an opportunity for future work, to evaluate the impact of differing terminology on real world antimicrobial use.

This study focused upon antibiotic use, given their prevalence in routine prescribing practices, however, as such it neglects other important areas of AMR including resistant fungal and parasitic infections. This approach is not intended to ignore these crucial areas, but to pave the way for a greater understanding of AMR within the general public, starting from a point where it appears easier to generate awareness, understanding and changes in behavioural intent.

There are a number of terminologies used to describe AMR that were not included in the intervention arms, notably 'Drug-Resistant Infections'. This reflects our co-design process with members of the public, and the fact that these terms did not score highly or resonate well with our public participants. Additionally, the co-design process used to create the study arms may have limited the novelty of the arms tested, as participants were asked to choose a term that resonated best with them, rather than one which had not been used before.

Our study focused upon the 'name' used to present AMR to the participant group, and this was the only differing factor between intervention arms. We acknowledge that there was information presented with each name that could have influenced the participants' comprehension of AMR, however, given that this was the same for each arm, whilst it may have influenced the overall scores (for example, not requesting antibiotics when unwell, or general comprehension), the impact of the different terminology to describe AMR will persist between arms.

This is an online study, and as such, participant data may reflect idealised responses rather than a true reflection of behavioural intent. As such, translation of our results into real world applications should be done with this in mind, as the impact of differing terminologies may not be the same.

Finally, this study was conducted in the United Kingdom, with a public engagement and co-design programme based in North West London. As such, the terminology selected for the intervention arms may not be reflective of the wider UK population or global opinions. Some of the terminology included, particularly 'Superbugs', may not translate well to a global audience.

There are a number of opportunities for future work. AMR is a global issue, and its presentation to the public must resonate with a large range of communities and populations. As such, it would be prudent to repeat our public engagement and co-design process to determine which terminology best conveys understanding of AMR and influences behavioural intent in different global settings. It would also be of interest to repeat this study focusing purely upon novel descriptions of AMR.

**Table 4 | Results for secondary and exploratory outcomes of the impact of different terminology to name antimicrobial resistance upon comprehension, recall and behavioural intent**

| Outcome | | Terminology used to name Antimicrobial Resistance (AMR) | | | |
|---|---|---|---|---|---|
| | | Antimicrobial Resistance —AMR (Control) *Participant score (%), p/CI* | Superbugs *Participant score (%), p/CI* | Antibiotic Resistance *Participant score (%), p/CI* | The Antibiotic Crisis *Participant score (%), p/CI* |
| **Secondary Outcome** | | | | | |
| Comprehension | *Correct answer[a]* | 63.3% | 65.1% $p = 0.069$, CI = [63.2%, 67.0%] | 66.0% $p = 0.013$, CI = [64.1%, 67.9%] | 65.4% $p = 0.058$, CI = [63.4%, 67.3%] |
| | *Confidence in answer* | 98.2% | 98.6% $p = 0.514$, CI = [97.2%, 99.3%] | 99.4% $p = 0.018$, CI = [98.5%, 99.7%] | 99.2% $p = 0.063$, CI = [98.1%, 99.6%] |
| **Exploratory Outcomes** | | | | | |
| Behavioural Intent | *Request Antibiotics* | 8.7% | 8.5% $p = 0.838$, CI = [6.3%, 11.3%] | 8.8% $p = 0.933$, CI = [6.5%, 11.8%] | 8.4% $p = 0.830$, CI = [6.2%, 11.3%] |
| | *Visit a GP* | 13.9% | 10.7% $p = 0.028$, CI = [8.3%, 13.5%] | 11.4% $p = 0.107$, CI = [8.9%, 14.5%] | 11.2% $p = 0.078$, CI = [8.7%, 14.3%] |
| | *Visit a Pharmacist* | 22.0% | 23.4% $p = 0.442$, CI = [19.9%, 27.3%] | 22.3% $p = 0.858$, CI = [18.9%, 26.2%] | 22.2% $p = 0.931$, CI = [18.7%, 26.0%] |
| | *Take other medications* | 40.6% | 38.5% $p = 0.331$, CI = [34.5%, 42.7%] | 43.6% $p = 0.160$, CI = [39.4%, 47.9%] | 42.1% $p = 0.494$, CI = [37.8%, 46.4%] |
| | *Stop antibiotics early (n = 376)* | 67.4% | 68.0% $p = 0.929$, CI = [53.0%, 80.0%] | 54.0% $p = 0.083$, CI = [38.3%, 69.0%] | 52.0% $p = 0.050$, CI = [36.2%, 67.4%] |
| Recall | | 72.0% | 65.4% $p = 0.001$, CI = [61.1%, 70.0%] | 79.0% $p = 0.0003$, CI = [75.4%, 82.2%] | 74.5% $p = 0.221$, CI = [70.1%, 78.0%] |

[a]*p* values in this row are adjusted for multiple comparisons using the Benjamini–Hochberg procedure. CIs are unadjusted.

Whilst our study demonstrated an impact of naming terminology on intent to finish a course of antibiotics, it did not influence intention to request antibiotics from a physician. Further work is required to understand the determinants of this behaviour, and what behaviourally informed interventions may be effective in reducing this.

Whilst feasible to conduct long-term follow up with our participants, and evaluate the relationship between initial comprehension and behavioural intent with sustained behaviour change, we are aware that re-contact studies often have a high attrition rate. However, a longitudinal study with repeated population level surveys would allow the impact of public health campaigns and different AMR terminology on attitudes, understanding and behavioural intent over time.

It would also be of benefit to evaluate our successful intervention arms —'Antibiotic Resistance' and 'The Antibiotic Crisis' in a real world setting, developing public communications strategies and evaluating the subsequent impact on antibiotic stewardship.

## Conclusion

This study adds to the building evidence that alternative terminology for AMR can be more effective in terms of public engagement and attitudes shift—the first step in engendering meaningful behaviour change. We demonstrate the benefits of co-creating such experiments with the public, allowing the study to focus on terms already deemed to be meaningful and resonant. We also highlight that work remains to be done on improving understanding of AMR in the general public, particularly when individuals are confident in their existing beliefs.

## Data availability

Analysed quantitative survey data is available within the manuscript, and full logistic regressions for all models are available in the Supplementary Data. Raw data is available at the project's Open Science Framework Page; https://osf.io/ne64z/?view_only=74417c7b1fc0421090ddb1ffcf5c6e37[26].

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

## Acknowledgements

This work received no specific grant from any funding agency, commercial or not-for-profit sectors. Public engagement activities were funded by Imperial College London Societal Engagement Seed Funding. The Fleming Initiative is jointly established by Imperial College London and Imperial College NHS Healthcare Trust.

## Author contributions

K.G.: Conceptualisation, Methodology, Investigation, Formal Analysis, Data Curation, Writing Original Draft, Visualisation, Project administration. A.F.: Methodology, Investigation, Formal Analysis, Data Curation, Writing—Original draft. B.S.: Methodology, Investigation, Formal Analysis, Supervision, Writing—review and editing. C.M.C.: Investigation, Writing—review and editing. A.J.: Formal Analysis, Data Curation, Writing—review and editing. E.M.Z.B.: Formal Analysis, Data Curation, Writing—review and editing. H.B.: Conceptualisation, Investigation, Writing—review and editing. S.H.: Conceptualisation, Writing—review and editing. H.B.: Conceptualisation, Investigation, Writing—review and editing. A.D.: Conceptualisation, Supervision, Writing—review and editing.

## Competing interests

The authors declare no competing interests.
