## [Peer Review file · Communications Medicine]

Terminology describing antimicrobial resistance influences attitudes, comprehension and behavioural intent. An online randomised behavioural experiment

Corresponding Author: Dr Kate Grailey

Version 0:

Reviewer comments:

Reviewer #1

(Remarks to the Author)

Many thanks for inviting me to review the article entitled: "Terminology describing antimicrobial resistance influences attitudes, comprehension and behavioural intent. An online randomised controlled trial", which I read with interest. The study presents a comparison of different AMR-related terms and their respective influences on lay audience's attitudes, knowledge and behavioural intention relevant to AMR. The authors show that "Antibiotic resistance" was most effective in creating positive intentions, but both Antibiotic resistance and Antibiotic Crisis reduced behavioural intentions to stop a course of antibiotics early.

I believe that the authors addressed an important topic. There was a clear study rationale and I commend the use of social science approaches to tackling to issue of AMR. Strengths of the study include the use of co-design elements and a large and representative sample.

Having said that, I have major concerns around the amount of detail provided regarding design, methods and analysis. I don't think the information contained within the manuscript is sufficient for enabling researchers to reproduce the work. For example, the "main measures section" contains very little detail and does not include information about how variables were computed (e.g., how was "recall" measured)? Also, the authors mention logistic regression models in their statistical analysis plan but then fail to report regression analyses in the Results section. More information on the intervention co-design would also have been helpful (including an overview of the total number of contributors and a demographic profile). I'm also missing a justification for some of the chosen variables. For example, I question the meaningfulness of some of the items designed to measure behavioural intent. I would argue that the intention to visit a GP or a pharmacist is not problematic in itself.

Additionally, I feel that the labelling of the study as an RCT misrepresents the nature of what is essentially a psychological experiment with random allocation to four different treatment conditions.

Overall, while an important research topic with some interesting results, I have concerns about the rigour of the research and would therefore recommend rejection of the manuscript.

Reviewer #2

(Remarks to the Author)

1. The study's primary claim is both novel and relevant given the significant need for improved public understanding of AMR and the behavioral changes necessary to address it. In my experience of working in the field of AMR awareness within community settings, the most struggled part was, the terminology. We need to make aware of something which is completely unknown to community people. Hence, I believe this study will contribute to future studies. Furthermore, this study could significantly impact public health communication by providing evidence-based guidance on how to frame AMR information. This has potential implications for policymakers, health organizations, and communication specialists. If successful, this work could lead to the development of widely adopted framing techniques to improve AMR awareness. This area is timely, as addressing AMR hinges on public understanding and engagement.

2. Strength of Evidence and Convincing Nature of the Work

- The randomized control trial (RCT) with a large sample size (4,000 participants) provides a strong framework for assessing the effects of different framing methods on AMR comprehension and attitudes.
- The structured approach to measure recall, comprehension, intent, and attitudes comprehensively covers the study's objectives.
- The inclusion of covariates like age, gender, region, ethnicity, and prior antibiotic use ensures a representative sample, which strengthens the study's generalizability to the UK population. I would suggest to add education as well.
- 'Blinding and Attention Checks' methodological aspects enhance the robustness of findings by minimizing potential bias and ensuring participant engagement.

3. Statistical Analysis and Reproducibility

- The use of quasibinomial regression and logistic regression models is appropriate given the binary nature of the primary and secondary outcomes. - - Adjusting for multiple comparisons using the Benjamini-Hochberg procedure for the primary and secondary outcomes adds rigor to the analysis.
- The study provides a detailed participant journey and survey structure, which enhances the reproducibility of the work. However, it would be beneficial if the authors clarified the criteria for "comprehension" and "intent" scoring within the survey, as this would support reproducibility and interpretation.
- While anonymized, securely stored data transfer to ICL is appropriate, transparency could be improved if the raw data, or at least de-identified datasets, were made publicly available to facilitate further analysis.

4. Further Questions and Considerations

- Would it be feasible to include additional long-term follow-up to assess if initial comprehension and intent influence sustained behavior change over time?
- The study's use of an online platform means results may reflect idealized responses rather than real-life behavior, especially in healthcare decision-making. The authors could consider discussing these limitations and how they might influence the translation of results into practical applications.

*****The transparency of the reviewing process is respected, and I am comfortable either signing this report or maintaining anonymity based on the authors' preferences.

Reviewer #3

(Remarks to the Author)

This manuscript sets out to test what types of terminology in the AMR/ABR field are effective in increasing awareness of the problem and inspiring changes in behaviour. This type of research has been requested by many in the field (including myself), and is thus very much welcomed. The co-designing approach unfortunately limits the novelty of the study, given that 3 out of 4 terms have already been studied. With this said, it must be noted that the reports from Wellcome Trusts are not in the realm of peer-reviewed scientific literature. I also understand that inclusion of more terms would increase the complexity of the study and decrease the chance to show results with any confidence, especially given the small differences shown already with these terms. Other terms would probably be best to leave to a second study, which could focus on more novel terms.

In general, the paper is well written (see minor remarks/questions below), of decent length and easy to understand also for non-behaviouralists. The paper discusses the findings and limitations well without overstating the findings.

Specific comments:

Introduction, last paragraph before study aims: The last sentence reads: "Consequently, much of the existing academic literature calls for new name or way to present AMR, with standardised terminology that is accessible to the public (22)." However, the paper referenced as no 22 was published earlier than references 18-21. While the reference supports the claim that literature calls for a new name or ways to communicate, it can not be described as "consequently" since it is temporally prior. The sentence should be reworded or the reference should be changed to a newer one.

Methods, study delivery: The methods are described in sufficient detail to be able to reproduce the study, although the manuscript does not describe the randomisation process. Were participants assigned to groups completely at random or was randomisation weighted in order to distribute the participants evenly based on demographics or prior knowledge? This should be stated.

Relating to outcomes and statistics, it is noted that the statistical analysis accounts for multiplicity. As a very minor comment however, I wonder if the secondary outcome measure should rather be a co-primary outcome. Please discuss this.

Results: Relating to the question above, participant characteristics should be presented based on assigned study groups to be able to assess potential bias or skewness of data.

Table 1 describes the demographics of participants. I would however like to understand what lies behind the parameter Education: Degree/No degree. Maybe this is country-specific, but it would be helpful to understand if this concerns a degree equivalent to a bachelor's, or something else. Also, it would be of interest to include whether the participants are healthcare professionals as this should be a decent predictor of knowledge and compliance to stewardship (based on the supplementary data, this information should be available to the authors). Third, the authors do not appear to have tested knowledge and attitudes at baseline. Knowledge at baseline could be a powerful effect modulator.

Discussion: The discussion is well written and organised, making it easy to follow. I consider previous data and some limitations of the study. I note that the reports from Wellcome Trusts have not been discussed here, e.g. noting that one of the WT reports raised concerns against the term "superbugs", which could be considered confirmed by this study - or at least help explain the performance of the term. While the discussion section does discuss previous studies, no quantitative

comparison is made, e.g. to Krockow in 2023 (ref 21) which also tested AMR against ABR (as discussed in the introduction. References: There appears to be a mistake in referencing: In the introduction, references 4 and 5 should be clustered with reference 6 at the end of the section, whereas reference 3 should be where 4 and 5 are currently.

Supplementary material: Comprehension question "What has caused the issue of Superbugs / Antibiotic Resistance / Antibiotic crisis / AMR". If only faced with the question in the context of the term Antibiotic crisis, and having prior knowledge, one could easily consider the option "No discovery of new antibiotics" since the term antibiotic crisis is rather broad. While I do not expect this to impact the results, it would be good for the authors to double-check that this is not the case.

Finally, two language comments. Please note I am not a native speaker, so these issues may not be technically incorrect.

1. The term "public members" gives me an association to "public figure" and raises the questions what they are members of. Consider rephrasing to e.g. "Members of the public".
2. In the introduction section, second paragraph, there is a clause "this itself determined by [...]" that I find difficult to understand please rephrase if possible.

Version 1:

Reviewer comments:

Reviewer #1

(Remarks to the Author)

Many thanks for inviting me to review the revised submission for this article. When I reviewed the original version, I had concerns about the rigour of the research, but these have been well addressed through the provision of the data set and detailed clarifications with regard to my methodological questions. I commend the authors for making those changes. My only remaining concern lies with the labelling of the study type as a randomised controlled trial. The authors have made an adjustment in that they clarified that this was an online experiment. However, I still find it misleading.

I am aware that different definitions of RCTs exist, but some include the following:

"Randomized controlled trials (RCT) are prospective studies that measure the effectiveness of a new intervention or treatment." (doi: 10.1111/1471-0528.15199)

"they all aim to obtain objective data to evaluate interventions with respect to an associated outcome in a target population." (<https://doi.org/10.1016/j.chest.2020.03.013>)

Definitions typically refer to the test of interventions or (medical) treatments. I don't think these criteria are fulfilled in this study, and I still believe the label of an (online) RCT is misleading. I would recommend removing this and simply calling it an online experiment.

Pending this final revision, I would recommend publication.

Reviewer #2

(Remarks to the Author)

The manuscript looks good to me. My comments and queries are well addressed.

Response to Reviewers

“Terminology describing antimicrobial resistance influences attitudes, comprehension and behavioural intent. An online randomised controlled trial”

We would like to thank the Editors and Reviewers for their thorough review of our manuscript. We have made edits to the manuscript to address these points, without significant alterations to the previous analyses or conclusions drawn. Please find our point-by-point response in line below.

Referee expertise:

Referee #1: AMR terminology, surveys, regression analysis

Referee #2: community attitudes to AMR, surveys

Referee #3: AMR, terminology, policy

Reviewers' comments:

Reviewer #1 (Remarks to the Author):

Many thanks for inviting me to review the article entitled: "Terminology describing antimicrobial resistance influences attitudes, comprehension and behavioural intent. An online randomised controlled trial", which I read with interest. The study presents a comparison of different AMR-related terms and their respective influences on lay audience's attitudes, knowledge and behavioural intention relevant to AMR. The authors show that "Antibiotic resistance" was most effective in creating positive intentions, but both Antibiotic resistance and Antibiotic Crisis reduced behavioural intentions to stop a course of antibiotics early.

I believe that the authors addressed an important topic. There was a clear study rationale and I commend the use of social science approaches to tackling the issue of AMR. Strengths of the study include the use of co-design elements and a large and representative sample.

Thank you for your review.

Having said that, I have major concerns around the amount of detail provided regarding design, methods and analysis. I don't think the information contained within the manuscript is sufficient for enabling researchers to reproduce the work. For example, the "main measures section" contains very little detail and does not include information about how variables were computed (e.g., how was "recall" measured)?

Thank you for highlighting this. We acknowledge that we did not include enough detail in our original submission, but we are confident that our revisions contain the requested detail. We have addressed the amount of detail regarding design, methods and analysis, with the intention of enabling researchers to reproduce the work. Within the methods, more detail regarding each of the main measures and how each variable was computed has been added, to support the information available with the full survey in supplementary file 1. We have also made the clean non-identifiable data available in an online repository - OSF to facilitate further analysis. https://osf.io/ne64z/?view_only=74417c7b1fc0421090ddb1ffcf5c6e37.

Also, the authors mention logistic regression models in their statistical analysis plan but then fail to report regression analyses in the Results section.

Thank you. We have now included the full regression tables within an additional supplementary file.

More information on the intervention co-design would also have been helpful (including an overview of the total number of contributors and a demographic profile).

Thank you for highlighting this. More detail on the process of intervention co-design has been added to the methods section of the manuscript, including the total number of contributors and the nature of each co-design session. As this was a public engagement activity, we did not record and store exact participant demographic information in line with GDPR, but an improved overview over the participant group has been provided.

I'm also missing a justification for some of the chosen variables. For example, I question the meaningfulness of some of the items designed to measure behavioural intent. I would argue that the intention to visit a GP or a pharmacist is not problematic in itself.

The intention with these questions was to evaluate behavioral intent with respect to antimicrobial stewardship. We agree, intention to visit a GP or pharmacist is not problematic in itself, however we wanted to explore the specificity of the intervention - for example, does one terminology make individuals more likely to visit the pharmacist for advice as they do not feel they require a prescription for antibiotics. We also felt it would be interesting to evaluate impact on visiting a GP - whilst this could be a good impact of the intervention (as above, knowledge that antibiotics not required and therefore an appointment not needed), it is also useful to understand the potential impact of different terminology on these behaviours, as there is also a risk that attitudes towards AMR may reduce the number of sick people visiting a Doctor, even if they need to. This question also provides rationale for future qualitative work to further understand the consequences of improving AMR education, attitudes and understanding in the general public.

This detail has been added to the methods and discussion.

Additionally, I feel that the labelling of the study as an RCT misrepresents the nature of what is essentially a psychological experiment with random allocation to four different treatment conditions.

Thank you for raising this. We feel that the design of this study is reasonably labelled as a randomised controlled trial, given it compares the effectiveness of treatments through the random assignment of participants to treatment arms, and measures the outcomes. The study also has a well defined control arm. We have reviewed the manuscript to ensure that it is clear the RCT is an online trial (rather than a real world field trial or clinical study).

Overall, while an important research topic with some interesting results, I have concerns about the rigour of the research and would therefore recommend rejection of the manuscript.

We have worked hard to address the points raised through your review and believe we have provided sufficient detail to allow replication of the study and to demonstrate the rigour of our work. Thank you for raising these - we believe this has allowed us to produce an improved manuscript, and increase the completeness of our reporting. We hope that you will reconsider your recommendation.

Reviewer #2 (Remarks to the Author):

1. The study's primary claim is both novel and relevant given the significant need for improved public understanding of AMR and the behavioral changes necessary to address it. In my experience of working in the field of AMR awareness within community settings, the most struggled part was, the terminology. We need to make aware of something which is completely unknown to community people. Hence, I believe this study will contribute to future studies. Furthermore, this study could significantly impact public health communication by providing evidence-based guidance on how to frame AMR information. This has potential implications for policymakers, health organizations, and communication specialists. If successful, this work could lead to the development of widely adopted framing techniques to improve AMR awareness. This area is timely, as addressing AMR hinges on public understanding and engagement.

Thank you for your review

2. Strength of Evidence and Convincing Nature of the Work

- The randomized control trial (RCT) with a large sample size (4,000 participants) provides a strong framework for assessing the effects of different framing methods on AMR comprehension and attitudes.

- The structured approach to measure recall, comprehension, intent, and attitudes comprehensively covers the study's objectives.

- The inclusion of covariates like age, gender, region, ethnicity, and prior antibiotic use ensures a representative sample, which strengthens the study's generalizability to the UK population. I would suggest to add education as well.

Thank you. We have verified, and can confirm that Education is already included as a covariate in all analyses. Education is listed as a covariate in the methods section of the manuscript.

- 'Blinding and Attention Checks' methodological aspects enhance the robustness of findings by minimizing potential bias and ensuring participant engagement.

3. Statistical Analysis and Reproducibility

- The use of quasibinomial regression and logistic regression models is appropriate given the binary nature of the primary and secondary outcomes. Adjusting for multiple comparisons using the Benjamini-Hochberg procedure for the primary and secondary outcomes adds rigor to the analysis.

- The study provides a detailed participant journey and survey structure, which enhances the reproducibility of the work. However, it would be beneficial if the authors clarified the criteria for "comprehension" and "intent" scoring within the survey, as this would support reproducibility and interpretation.

Thank you for raising this. We have added increased detail to the methods section to demonstrate how these variables were constructed and computed, supported by the full survey in supplementary file 1. We feel confident the manuscript now facilitates reproducibility of our work.

- While anonymized, securely stored data transfer to ICL is appropriate, transparency could be improved if the raw data, or at least de-identified datasets, were made publicly available to facilitate further analysis. **Thank you for highlighting this. We have made the de-identified datasets available in an online repository - the Open Science Framework.**
https://osf.io/ne64z/?view_only=74417c7b1fc0421090ddb1ffcf5c6e37

4. Further Questions and Considerations

- Would it be feasible to include additional long-term follow-up to assess if initial comprehension and intent influence sustained behavior change over time?

Thank you for raising this - this would be a valuable and interesting piece of work to conduct. It is feasible to run a "recontact" study, where we could follow up participants of this study and evaluate the impact of initial comprehension and intent on sustained behavioural intent, however, from experience the attrition rate for this type of work is very large, especially consider original data collection is now more than a few months in the past, and it may not be powered to show any significant findings.

However, it would be of benefit to conduct a longitudinal survey, perhaps at population level to measure comprehension and intent at regular intervals. This could be linked to real-world data on changes in behaviours such as rates of antimicrobial prescriptions.

This has been acknowledged in the discussion.

- The study's use of an online platform means results may reflect idealized responses rather than real-life behavior, especially in healthcare decision-making. The authors could consider discussing these limitations and how they might influence the translation of results into practical applications. **Thank you - this is a key point and we apologise for not including it earlier. We have added to the limitations section to acknowledge this.**

*****The transparency of the reviewing process is respected, and I am comfortable either signing this report or maintaining anonymity based on the authors' preferences.

Reviewer #3 (Remarks to the Author):

This manuscript sets out to test what types of terminology in the AMR/ABR field are effective in increasing awareness of the problem and inspiring changes in behaviour. This type of research has been requested by many in the field (including myself), and is thus very much welcomed. The co-designing approach unfortunately limits the novelty of the study, given that 3 out of 4 terms have already been studied. With this said, it must be noted that the reports from Wellcome Trusts are not in the realm of peer-reviewed scientific literature. I also understand that inclusion of more terms would increase the complexity of the study and decrease the chance to show results with any confidence, especially given the small differences shown already with these terms. Other terms would probably be best to leave to a second study, which could focus on more novel terms.

Thank you. We acknowledge that the co-design process with members of the public may have limited novelty, and have added this to the limitations of the study within the manuscript.

In general, the paper is well written (see minor remarks/questions below), of decent length and easy to understand also for non-behaviouralists. The paper discusses the findings and limitations well without overstating the findings.

Thank you for your review.

Specific comments:

Introduction, last paragraph before study aims: The last sentence reads: "Consequently, much of the existing academic literature calls for new name or way to present AMR, with standardised terminology that is accessible to the public (22)." However, the paper referenced as no 22 was published earlier than references 18-21. While the reference supports the claim that literature calls for a new name or ways to communicate, it can not be described as "consequently" since it is temporally prior. The sentence should be reworded or the reference should be changed to a newer one.

Thank you for highlighting this. The sentence has been reworded.

Methods, study delivery: The methods are described in sufficient detail to be able to reproduce the study, although the manuscript does not describe the randomisation process. Were participants assigned to groups completely at random or was randomisation weighted in order to distribute the participants evenly based on demographics or prior knowledge? This should be stated.

Thank you. The participants were assigned to groups completely at random. This has now been added to the methods section of the manuscript.

Relating to outcomes and statistics, it is noted that the statistical analysis accounts for multiplicity. As a very minor comment however, I wonder if the secondary outcome measure should rather be a co-primary outcome. Please discuss this.

Thank you for raising this interesting point, which we have discussed at length as a team. Given the analysis and outcome measures were pre-planned, we have elected to leave them as originally categorised, rather than re-categorising after seeing the results. However, we acknowledge that having it as a co-primary outcome and conducting multiple comparison correction could make this more rigorous.

Results: Relating to the question above, participant characteristics should be presented based on assigned study groups to be able to assess potential bias or skewness of data.

Thank you - the demographic has been reproduced according to the treatment arm.

Table 1 describes the demographics of participants. I would however like to understand what lies behind the parameter Education: Degree/No degree. Maybe this is country-specific, but it would be helpful to understand if this concerns a degree equivalent to a bachelor's, or something else.

Thank you - this parameter refers to a bachelors degree. This has been clarified in the manuscript.

Also, it would be of interest to include whether the participants are healthcare professionals as this should be a decent predictor of knowledge and compliance to stewardship (based on the supplementary data, this information should be available to the authors).

This has been added to the demographics tables - both overall demographics and by treatment demographics.

Third, the authors do not appear to have tested knowledge and attitudes at baseline. Knowledge at baseline could be a powerful effect modulator.

We agree that knowledge and attitudes might impact the treatment effect. This was not included in the study, as we did not measure baseline knowledge and attitudes to avoid priming participants. We remain confident that the effect estimate itself is robust to this considering this was an RCT and the characteristics of participants should be the same across arms (as we have also been able to confirm through the demographics tables).

Discussion: The discussion is well written and organised, making it easy to follow. I considers previous data and some limitations of the study. I note that the reports from Wellcome Trusts have not been discussed here, e.g. noting that one of the WT reports raised concerns against the term "superbugs", which could be considered confirmed by this study - or at least help explain the performance of the term.

Thank you - we have included this detail in the discussion section

While the discussion section does discuss previous studies, no quantitative comparison is made, e.g. to Krockow in 2023 (ref 21) which also tested AMR against ABR (as discussed in the introduction).

Thank you for highlighting this important omission - we have now included comparison to the 2023 Krockow et al work, exploring similarities and differences in the findings.

References: There appears to be a mistake in referencing: In the introduction, references 4 and 5 should be clustered with reference 6 at the end of the section, whereas reference 3 should be where 4 and 5 are currently.

Thank you for spotting this. The references have been amended accordingly and should now be accurate.

Supplementary material: Comprehension question "What has caused the issue of Superbugs / Antibiotic Resistance / Antibiotic crisis / AMR". If only faced with the question in the context of the term Antibiotic crisis, and having prior knowledge, one could easily consider the option "No

discovery of new antibiotics" since the term antibiotic crisis is rather broad. While I do not expect this to impact the results, it would be good for the authors to double-check that this is not the case.

We agree this is a possibility regarding comprehension of term itself, but not in terms of the stimulus we showed them. We have looked at the relationship between treatment arm and responses to this specific response, and did not find any evidence that the Antibiotic Crisis arm responded any differently. Thank you for raising this important point, and it was useful for us to review this.

Finally, two language comments. Please note I am not a native speaker, so these issues may not be technically incorrect.

1. The term "public members" gives me an association to "public figure" and raises the questions what they are members of. Consider rephrasing to e.g. "Members of the public".
2. In the introduction section, second paragraph, there is a clause "this itself determined by [...]" that I find difficult to understand please rephrase if possible.

Thank you - these two language comments have been addressed in the manuscript.

Response to Reviewers

“Terminology describing antimicrobial resistance influences attitudes, comprehension and behavioural intent. An online randomised controlled trial”

We would like to thank the Editors and Reviewers for their thorough review of our manuscript. We have made edits to the manuscript to address these points, without significant alterations to the previous analyses or conclusions drawn. Please find our point-by-point response in line below.

Reviewer #1 (Remarks to the Author):

Many thanks for inviting me to review the revised submission for this article. When I reviewed the original version, I had concerns about the rigour of the research, but these have been well addressed through the provision of the data set and detailed clarifications with regard to my methodological questions. I commend the authors for making those changes. My only remaining concern lies with the labelling of the study type as a randomised controlled trial. The authors have made an adjustment in that they clarified that this was an online experiment. However, I still find it misleading.

I am aware that different definitions of RCTs exist, but some include the following:

"Randomized controlled trials (RCT) are prospective studies that measure the effectiveness of a new intervention or treatment." (doi: 10.1111/1471-0528.15199)

"they all aim to obtain objective data to evaluate interventions with respect to an associated outcome in a target population." (<https://doi.org/10.1016/j.chest.2020.03.013>)

Definitions typically refer to the test of interventions or (medical) treatments. I don't think these criteria are fulfilled in this study, and I still believe the label of an (online) RCT is misleading. I would recommend removing this and simply calling it an online experiment.

Pending this final revision, I would recommend publication.

Thank you for your review. We have adjusted the way the study is labelled to an 'online randomised behavioural experiment', and removed any mention of a trial.

Reviewer #2 (Remarks to the Author):

The manuscript looks good to me. My comments and queries are well addressed.

Thank you for your review.